# Geographic weighted regression analysis of hot spots of anemia and its associated factors among children aged 6–59 months in Ethiopia: A geographic weighted regression analysis and multilevel robust Poisson regression analysis

**Getayeneh Antehunegn Tesema**[ID]*, **Zemenu Tadesse Tessema, Dessie Abebaw Angaw, Koku Sisay Tamirat, Achamyeleh Birhanu Teshale**

Department of Epidemiology and Biostatistics, Institute of Public Health, College of Medicine and Health Sciences, University of Gondar, Gondar, Ethiopia

* getayenehantehunegn@gmail.com

## Abstract

### Introduction

Anemia among children aged 6–59 months remains a major public health problem in low-and high-income countries including Ethiopia. Anemia is associated with significant consequences on the health of children such as under-five morbidity and mortality, increased risk of infection, and poor academic performance. The prevalence of anemia in Ethiopia has varied across areas. Therefore, this study aimed to investigate the geographic weighted regression analysis of anemia and its associated factors among children aged 6–59 months in Ethiopia.

### Methods

This study was based on the 2016 Ethiopian Demographic and Health Survey (EDHS) data. A total weighted sample of 8482 children aged 6–59 months was included. For the spatial analysis, Arc-GIS version 10.7 and SaTScan version 9.6 statistical software were used. Spatial regression was done to identify factors associated with the hotspots of anemia and model comparison was based on adjusted $R^2$ and Corrected Akaike Information Criteria (AICc). For the associated factors, the multilevel robust Poisson regression was fitted since the prevalence of anemia was greater than 10%. Variables with a p-value < 0.2 in the bi-variable analysis were considered for the multivariable analysis. In the multivariable multilevel robust Poisson regression analysis, the adjusted prevalence ratio with the 95% confidence interval was reported to declare the statistical significance and strength of association.

### Results

The prevalence of anemia among children aged 6–59 months was 57.56% (95%CI: 56.50%, 58.61%) with significant spatial variation across regions in Ethiopia. The significant

**Data Availability Statement:** The data we used for this study was EDHS 2016 data, which is publicly

available in the measure DHS program. We accessed this data after explaining the purpose of this study and therefore, everybody can access this data from this https://dhsprogram.com/data/ link.

**Funding:** The author(s) received no specific funding for this work.

**Competing interests:** The authors have declared that no competing interests exist.

**Abbreviations:** APR, Adjusted Prevalence Ratio; CI, Confidence Interval; CPR, Crude Prevalence Ratio; EDHS, Ethiopian Demographic and Health Survey; ICC, Intra-class Correlation Coefficient; LLR, log-likelihood Ratio; LR, Likelihood Ratio; SSA, Sub-Saharan Africa; WHO, World Health Organization.

hot spot areas of anemia among children aged 6–59 months were detected in the central, west, and east Afar, Somali, Dire Dawa, Harari, and northwest Gambella regions. Mothers who had anemia, a child aged 23–59 months, mothers aged 15–19 years, and coming from a household with a poorer or poorest household were significant predictors of the spatial variations of anemia among children aged 6–59 months. In the multilevel robust Poisson analysis, born to mothers aged 30–39 (APR = 0.84, 95% CI: 0.76, 0.92) and 40–49 years (APR = 0.73, 95% CI: 0.65, 0.83), mothers who didn't have formal education (APR = 1.10, 95% CI: 1.00, 1.20), Children in the poorest household wealth index (APR = 1.17, 95% CI: 1.06, 1.29), being 4–6 (APR = 1.08, 95% CI: 1.02, 1.13) and above 6 order of birth (APR = 1.15, 95% CI: 1.07, 1.23), children born to anemic mothers (APR = 1.24, 95% CI: 1.19, 1.29), children aged 24–59 months (APR = 0.70, 95% CI: 0.68, 0.73), stunted children (APR = 1.09, 95% CI: 1.04, 1.13) and underweight children (APR = 1.07, 95% CI: 1.03, 1.13) were significantly associated with anemia among children aged 6–59 months.

## Conclusion and recommendation

Anemia is still a public health problem for children in Ethiopia. Residing in a geographic area where a high proportion of children born to mothers aged 15–19 years, a child aged 6–23 months, coming from a household with poorer or poorest wealth index, and mothers with anemia increased the risk of experiencing anemia among children aged 6–59 months. Maternal education, maternal age, child age, household wealth, stunting, underweight, birth order, and maternal anemia were significant predictors of anemia among children. The detailed map of anemia hot spots among children aged 6–59 months and its predictors could assist program planners and decision-makers to design targeted public health interventions.

## Background

Anemia is the commonest public health problem in Low-and Middle-Income Countries (LMICs) [1–3]. Globally, an estimated 1.62 billion people are anemic, of these more than 43% occurred in LMICs particularly in Asia and Africa [4, 5]. Under-five children are highly affected by anemia compared to the general population [6, 7]. According to World Health Organization (WHO), an estimated 293 million under-five children are anemic globally with a prevalence of 47.4% [8]. In Ethiopia, the prevalence of anemia among under-five children was 57% [7, 9]. Anemia during childhood has been linked to developmental delay, recurrent infections, reduced working capacity, poor school performance, and dilated cardiomyopathy [10, 11].

Anemia has a multifactorial etiology and several factors act simultaneously. Nutritional deficiencies such as iron, folate [12, 13], vitamin B12 [14], and vitamin A [15] are the leading causes of anemia among under-five children in LMICs. Besides, diseases like malaria [16], Visceral Leishmaniasis (VL) [17], hookworm [18], schistosomiasis [19], cancer [20], tuberculosis [21], HIV/AIDS [22], and genetic hemoglobinopathies are commonly reported causes of anemia in developing and developed world [23].

Previous studies on anemia among under-five children revealed that residence, maternal educational status, taking drugs for intestinal parasites, child nutrition status (stunting,

wasting, and underweight), maternal age, child age, household wealth index, maternal anemia status, child-size at birth, birth order, parity, type of water source, type of toilet facility, type of birth, sex of children, and media exposure were significantly associated with anemia [24–27].

According to studies reported on the prevalence and associated factors of anemia, the prevalence of anemia has significant variation across regions in Ethiopia [7, 28–34]. Therefore, this study aimed to investigate spatial regression analysis of anemia and its associated factors among children aged 6–59 months. The findings of this study will help policymakers, program planners, and other health care programs in guiding health programs and prioritize prevention and intervention programs. Besides, mapping hotspot areas of anemia will provide a deeper understanding of the impacts of already implemented interventions in each region of the country.

## Methods and materials

### Study area, data source, and study period

This study was based on the 2016 Ethiopian Demographic and Health Survey (EDHS) data. The 2016 EDHS was the fourth DHS in Ethiopia, which was conducted every 5 years. The EDHS is mainly aimed to generate updated health and health-related indicators such as maternal mortality, child mortality, family planning, vaccination, and maternal health care service utilization. Ethiopia is administratively divided into nine geographical regions (Tigray, Afar, Amhara, Oromia, Somalia, Benishangul-Gumuz, Southern Nation Nationality, and People's Region (SNNPR), Gambella, and Harari) and two self-administrative cities (Addis Ababa and Dire Dawa). Each region is subdivided into zones, each zone into Weredas, and each Wereda into Kebeles (which is the lowest administrative unit). A multistage stratified sampling technique was applied to select the study designs. In the first Enumeration Areas (EAs) was randomly selected and in the second on average 28 households per clusters/EAs.

### Sample and population

Hemoglobin testing was carried out among children aged 6–59 months in the selected households using HemoCue rapid testing methodology. For the test, a drop of capillary blood was taken from a child's fingertip or heel and was drawn into the micro cuvette which was then analyzed using the photometer that displays the hemoglobin concentration. Then, anemic status was determined based on the hemoglobin level. The EDHS has several datasets such as men (MR file), women (IR file), children (KR file), birth (BR file), and household (HR file) datasets. For this study, we used the Kids Record dataset (KR file), and a total weighted sample of 8482 children aged 6–59 months was included.

### Study variables

The dependent variable was the anemia status of children aged 6–59 months, which was categorized into anemic (hemoglobin level < 11 g/dl) and not anemic (hemoglobin $\geq$11.0 g/dl). It was assessed based on the hemoglobin concentration in blood adjusted for altitude.

In EDHS, before determining a child is anemic or not, hemoglobin adjustment for altitude was done by subtracting or adding the adjusted Hgb value to each individual observed Hgb value.

The Hgb adjustment was made using the formula;

$$\text{Hgb adjustment} = -0.0322 \, (\text{altitude} * 0.0032808) + 0.022 \, (\text{altitude} * 00032808)^2$$

The adjustment for altitude was done to take into account the reduction in oxygen saturation of the blood. The independent variables considered in this study were region, residence, maternal age, maternal educational status, household wealth status, media exposure, maternal anemia status, sex of children, type of birth, age of children, size of child at birth, water source, type of toilet facility, parity, birth order, taking drugs for intestinal parasites in the last 6 months, wasting status (Z-scores for Weight-for-Height (WHZ)), underweight status (Z-scores for Weight-for-Age (WAZ)) and stunting status (Z-scores for Height-for-Age (HAZ)).

Stunting is defined as children with height-for-age Z-score (HAZ) $<-2SD$, wasting is defined as children with weight-for-height Z-score (WHZ) $<-2SD$, and underweight is defined as children with weight-for-age Z-score (WAZ) $<-2SD$. Maternal anemia was defined as "mild", "moderate", and "severe anemia" when Hgb level ranges 10–10.9 g/dl, 7–9.9 g/dl, and $<7$ g/dl, respectively.

## Data management and analysis

**Factors associated with anemia.** Data extraction, coding, and analysis were done using Stata version 14 and Arc-GIS version 10.6 statistical software. The weighted data were used for analysis to restore the representativeness of the data. Since the EDHS data has a hierarchical nature, the Intra-class Correlation Coefficient (ICC) was estimated to assess the clustering effect. The ICC indicated that there was a significant clustering effect (ICC>10%). This study was a cross-sectional study and the prevalence of anemia was greater than 10%, and if we reported the odds ratio it could overestimate the association between anemia and the independent variables. In such cases, the prevalence ratio is the best measure of association, and therefore, multilevel Poisson regression analysis with robust variance was fitted to identify predictors of anemia. Variables with a p-value<0.2 in the bi-variable multilevel Poisson regression analysis were considered for the multivariable analysis. Deviance was used to verify model fitness, and a model with the lowest deviance was considered the best-fit model. Finally, the Adjusted Prevalence Ratio (APR) with its 95% confidence interval (CI) was reported, and variables with p value<0.05 in the multivariable analysis were considered as significant predictors of anemia among under-five children.

**Spatial analysis.** The global spatial autocorrelation (Global Moran's I) was done to assess whether the spatial distribution of anemia among under-five children in Ethiopia was dispersed, clustered, or randomly distributed [35]. Global Moran's I is a spatial statistic used to measure spatial autocorrelation by taking the entire data set and produce a single output value that ranges from -1 to +1. Moran's, I value close to −1 indicates that anemia among under-five children is dispersed, whereas Moran's I close to +1 indicates anemia among under-five children is clustered and if Moran's I close to 0 revealed that anemia among under-five children is randomly distributed. A statistically significant Moran's I (p < 0.05) value showed that anemia among under-five children is non-random. The hotspot analysis was done using the Getis-OrdGi* statistics to explore how spatial autocorrelation varies over the study location by calculating GI* statistic for each area. Z-score is computed to determine the statistical significance of clustering, and the p-value is computed for the significance. Statistical output with high GI* indicates "hotspot" whereas low GI* means a "cold spot" [36].

**Spatial regression analysis.** The Ordinary Least Square (OLS) regression and Geographic Weighted Regression (GWR) statistical analysis were employed for exploring the spatial relationship between anemia among under-five children and the explanatory variables. The outcome variable for spatial regression analysis was the percentage of anemia among under-five children at the EA level. A neighborhood or bandwidth is the distance band or the number of neighbors used for each regression equation, it is the most important parameter for spatial

regression as it controls the degree of smoothening in the model. The complexity of spatial regression model depends not only by the number of variables in the model but also the bandwidth. There are three choice of band width methods such as AICc, CV and bandwidth parameter. For this study we have used adaptive kernel whose bandwidth was found by minimizing the AICc value.

*Ordinary Least Squares (OLS) regression.* The spatial regression modeling was performed to identify predictors of the spatial heterogeneity of anemia among under-five children. OLS is a global statistical model for testing and explaining the relationship between the dependent and independent variables [37]. It uses a single equation to estimate the relationship between the dependent and independent variables and assumes stationarity or consistent relationship across the study area. The OLS was used as a diagnostic tool and for selecting the appropriate predictors (concerning their relationship with anemia) for the Geographic Weighted Regression (GWR) model [38].

The OLS can automatically check the multicollinearity between independent variables (redundancy among explanatory variables). The multicollinearity was assessed using the Variance Inflation Factor (VIF). If the VIF values are greater than 10 in the OLS model, it indicates the existence of multicollinearity among the explanatory variables and should apply to leave one out an approach based on the VIF values. Besides, the autocorrelation statistic was applied to detect whether there is spatial autocorrelation or clustering of the residuals which violates the assumptions of OLS. The spatial independence of the residuals was assessed with the global spatial autocorrelation coefficient Moran's I value. The Moran's I value ranges from +1 (positive autocorrelation) and -1 (negative autocorrelation).

*Geographically Weighted Regression (GWR).* A local spatial statistical technique that assumes the non-stationarity in relationships/ heterogeneity in the relationship between the dependent and explanatory variables across EAs [38–40]. The GWR analysis is considered when the Koenker statistics is significant (p-value<0.05), which means the relationships between the dependent and the independent variable change from location to location. In the GWR analysis, the coefficients of the explanatory variables take different values across the study area. Mapping the GWR coefficients associated with the explanatory variables, which are produced using the GWR, provides insight for targeted interventions. The corrected Akaike Information Criteria (AICc) and adjusted R-squared for model comparison of OLS (global model) and GWR (local) model. A model with the lowest AICc value and a higher adjusted R-squared value was considered as the best-fitted model for the data.

## Results

### Descriptive results

A total weighted sample of 8482 children aged 6 to 59 months was included. Of these, more than half (51.85%) of the children were males. The majority (43.87%) of the children were in the Oromia region and 7621 (89.85%) were from the rural areas. Nearly half of the mothers (48.76%) were aged 20–29 years, and 5684 (67.02%) of the mothers didn't attain formal education. About 1988 (23.44%) and 1142 (13.46%) of the mothers fall within the poorest and richest household index quintiles, respectively. Nearly one-third (30.1%) of the children's mothers were anemic. Regarding children's nutritional status, about 40.73%, 25.29%, and 9.38% of the children were stunted, underweight and wasted, respectively (Table 1).

### Prevalence of anemia among under-five children in Ethiopia

The prevalence of anemia among under-five children was 57.56% (95%CI: 56.50%, 58.61%). The highest prevalence of anemia among under-five children was observed in Somali

**Table 1. Descriptive characteristics of the study participants in Ethiopia, 2016.**

| Variables | Weighted frequency | Percentage |
|---|---|---|
| **Region** | | |
| Tigray | 572 | 6.75 |
| Afar | 83 | 1.00 |
| Amhara | 1657 | 19.54 |
| Oromia | 3722 | 43.87 |
| Somali | 349 | 4.11 |
| Benishangul-gumuz | 90 | 1.07 |
| SNNPRs | 1781 | 20.99 |
| Gambella | 20 | 0.23 |
| Harari | 16 | 0.19 |
| Addis Ababa | 161 | 1.90 |
| Dire-Dawa | 31 | 0.37 |
| **Residence** | | |
| Rural | 7621 | 89.85 |
| Urban | 861 | 10.15 |
| **Maternal age (years)** | | |
| <20 | 226 | 2.67 |
| 20–29 | 4136 | 48.76 |
| 30–39 | 3335 | 39.31 |
| ≥40 | 785 | 9.25 |
| **Maternal education status** | | |
| No | 5684 | 67.02 |
| Primary | 2281 | 26.89 |
| Secondary or higher | 516 | 6.09 |
| **Household wealth status** | | |
| Poorest | 1988 | 23.44 |
| Poorer | 1989 | 23.45 |
| Middle | 1823 | 21.49 |
| Richer | 1540 | 18.15 |
| Richest | 1142 | 13.46 |
| **Media exposure** | | |
| No | 5787 | 68.22 |
| Yes | 2695 | 31.78 |
| **Maternal anemia status** | | |
| Not anemic | 5861 | 69.90 |
| Anemic | 2524 | 30.10 |
| **Sex of children** | | |
| Male | 4398 | 51.85 |
| Female | 4084 | 48.15 |
| **Type of birth** | | |
| Single | 8274 | 97.55 |
| Multiple | 208 | 2.45 |
| **Age of children (months)** | | |
| 6–23 | 2908 | 34.28 |
| 24–59 | 5574 | 65.72 |
| **Size of children at birth** | | |
| Small | 2177 | 25.66 |

(*Continued*)

**Table 1.** (Continued)

| Variables | Weighted frequency | Percentage |
|---|---|---|
| Average | 3565 | 42.03 |
| Large | 2740 | 32.31 |
| **Water source** | | |
| Not improved | 3841 | 45.28 |
| Improved | 4641 | 54.71 |
| **Toilet facility** | | |
| Not improved | 3297 | 38.87 |
| Improved | 5185 | 61.13 |
| **Parity** | | |
| 1–3 | 3640 | 42.91 |
| 4–6 | 2945 | 34.72 |
| >6 | 1897 | 22.37 |
| **Birth order** | | |
| 1–3 | 4118 | 48.55 |
| 4–6 | 2816 | 33.20 |
| >6 | 1548 | 18.24 |
| **Taking drugs for intestinal parasites in the last 6 months** | | |
| No | 7391 | 87.14 |
| Yes | 1091 | 12.86 |
| **Stunting status** | | |
| Not stunted | 5027 | 59.27 |
| Stunted | 3455 | 40.73 |
| **Underweight** | | |
| No | 6337 | 74.71 |
| Yes | 2145 | 25.29 |
| **Wasting status** | | |
| Not wasted | 7687 | 90.62 |
| Wasted | 795 | 9.38 |

(83.24%), Afar (74.72%), and Dire Dawa (72.12%) regions. Whereas, the lowest prevalence of anemia was observed in Addis Ababa (48.71%), Benishangul-gumuz (43.20%), and Amhara (42.66%) regions (Fig 1).

## Factors associated with anemia among under-five children

**Random effect analysis results.** In the null model, the ICC value was 18.8% (95% CI: 15.99%, 21.98%), indicated that about 18.8% of the overall variability in anemia was explained by the between cluster variation while the remaining 81.20% was attributed to the individual-level variation. Besides, the Likelihood Ratio (LR) test was (LR test vs. logistic model: $X^2(01) = 512.36$, $p < 0.0001$), which showed that the mixed-effect models were the best-fitted model for this data compared to the standard model.

**Fixed effect analysis results.** In the multivariable mixed-effect Poisson regression with a robust variance; maternal age, maternal education, household wealth index, birth order, maternal anemia, age of children, stunting, and underweight were significantly associated with anemia among under-five children. The prevalence of anemia among children born to mothers aged 30–39 and 40–49 years was decreased by 16% (APR = 0.84, 95% CI: 0.76, 0.92) and 27% (APR = 0.73, 95% CI: 0.65, 0.83) compared to children born to mothers aged less than 20

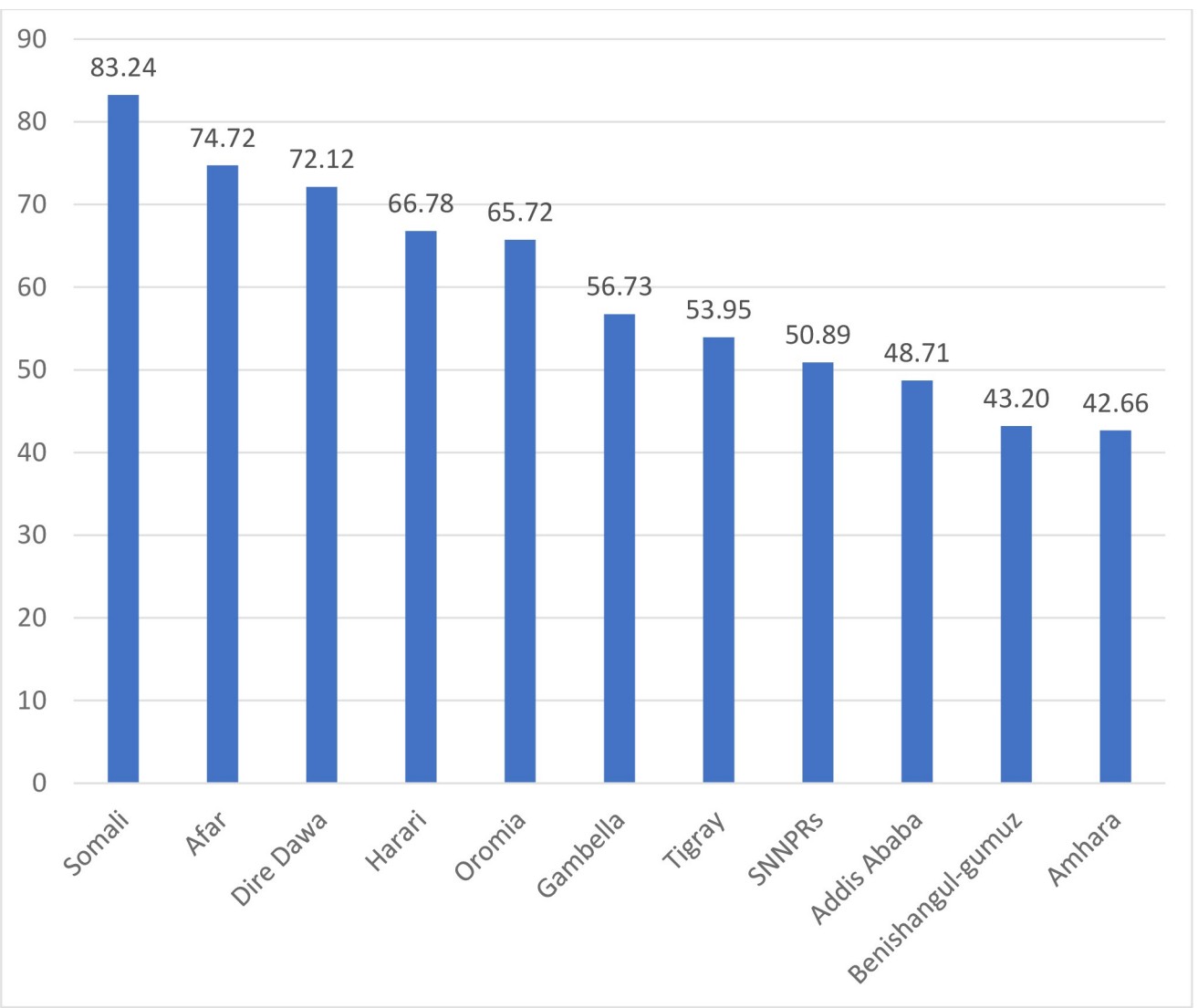

**Fig 1. The prevalence of anemia among under-five children across regions in Ethiopia, 2016.**

years, respectively. Children whose mothers did not have formal education had 1.10 times (APR = 1.10, 95% CI: 1.00, 1.20) higher prevalence of anemia than children whose mothers attained secondary education or higher. Children in the poorest household wealth index were 1.17 times (APR = 1.17, 95% CI: 1.06, 1.29) higher prevalence of anemia compared to children in the richest household wealth index. Being the 4th-6th and above 6th order of birth increases the prevalence of anemia by 1.08 (APR = 1.08, 95% CI: 1.02, 1.13) and 1.15 (APR = 1.15, 95% CI: 1.07, 1.23) than first to third births. Children born to anemic mothers had 1.24 times (APR = 1.24, 95% CI: 1.19, 1.29) a higher prevalence of anemia compared to children born to non-anemic mothers. The prevalence of anemia among children aged 24–59 months was decreased by 30% (APR = 0.70, 95% CI: 0.68, 0.73) compared to children aged 6–23 months. Stunted children had 1.09 times (APR = 1.09, 95% CI: 1.04, 1.13) a higher prevalence of anemia compared to no-stunted children, and underweight children had 1.07 times (APR = 1.07, 95% CI: 1.03, 1.13) higher prevalence of anemia compared to normal children (Table 2).

**Table 2. Bi-variable and multivariable mixed-effect robust Poisson regression analysis of anemia among under-five children in Ethiopia, 2016.**

| Variables | Anemia status | | Crude Prevalence Ratio with 95% CI | Adjusted Prevalence Ratio with 95% CI |
|---|---|---|---|---|
| | No | Yes | | |
| **Residence** | | | | |
| Urban | 608 | 720 | 1 | 1 |
| Rural | 2496 | 3971 | 1.47 (1.20, 1.79) | 0.95 (0.86, 1.04) |
| **Maternal age** | | | | |
| <20 | 61 | 169 | 1 | 1 |
| 20–29 | 1471 | 2411 | 0.59 (0.43, 0.81) | 0.93 (0.85, 1.01) |
| 30–39 | 1248 | 1772 | 0.53 (0.38, 0.73) | 0.84 (0.76, 0.92) * |
| 40–49 | 324 | 339 | 0.40 (0.28, 0.56) | 0.73 (0.65, 0.83)** |
| **Maternal educational status** | | | | |
| No | 1903 | 3181 | 1.44 (1.17, 1.76) | 1.10 (1.00, 1.20)* |
| Primary | 840 | 1141 | 1.29 (1.05, 1.60) | 1.08 (0.98, 1.18) |
| Secondary or above | 361 | 368 | 1 | 1 |
| **Household wealth index** | | | | |
| Poorest | 849 | 2005 | 2.17 (1.80, 2.60) | 1.17 (1.06, 1.29)** |
| Poorer | 568 | 819 | 1.50 (1.23, 1.82) | 1.03 (0.94, 1.14) |
| Middle | 543 | 613 | 1.22 (0.98, 1.49) | 0.96 (0.87, 1.07) |
| Richer | 465 | 518 | 1.22 (0.99, 1.50) | 0.98 (0.89, 1.09) |
| Richest | 679 | 736 | 1 | 1 |
| **Media exposure** | | | | |
| No | 1974 | 3356 | 1 | 1 |
| Yes | 1130 | 1335 | 0.73 (0.64, 0.82) | 0.96 (0.91, 1.01) |
| **Birth order** | | | | |
| 1–3 | 1652 | 2326 | 1 | 1 |
| 4–6 | 972 | 1557 | 1.09 (0.98, 1.23) | 1.08 (1.02, 1.13)** |
| >6 | 480 | 808 | 1.12 (0.96, 1.29) | 1.15 (1.07, 1.23)** |
| **Maternal anemia status** | | | | |
| Not anemic | 2291 | 2709 | 1 | 1 |
| Anemic | 781 | 1908 | 1.61 (1.43, 1.81) | 1.24 (1.19, 1.29)* |
| **Child size at birth** | | | | |
| Small | 735 | 1372 | 1 | 1 |
| Average | 1372 | 1944 | 0.85 (0.75, 0.96) | 0.96 (0.92, 1.01) |
| Large | 997 | 1375 | 0.80 (0.70, 0.92) | 0.97 (0.92, 1.02) |
| **Water source** | | | | |
| Not improved | 1160 | 2051 | 1 | 1 |
| Improved | 1944 | 2640 | 0.82 (0.72, 0.93) | 0.99 (0.95, 1.04) |
| **Type of toilet facility** | | | | |
| Not improved | 1195 | 2369 | 1 | 1 |
| Improved | 1909 | 2322 | 0.69(0.61, 0.78) | 0.96 (0.91, 1.01) |
| **Age of children (months)** | | | | |
| 6–23 | 3031 | 4578 | 1 | 1 |
| 24–59 | 73 | 113 | 0.33 (0.29, 0.37) | 0.70 (0.68, 0.73)* |
| **Taking drugs for intestinal parasites in the last 6 months** | | | | |
| No | 2621 | 4133 | 1 | 1 |
| Yes | 483 | 558 | 0.81 (0.70, 0.95) | 0.98 (0.92, 1.05) |
| **Stunting status** | | | | |
| Not stunted | 2014 | 2759 | 1 | 1 |

(*Continued*)

**Table 2.** (Continued)

| Variables | Anemia status | | Crude Prevalence Ratio with 95% CI | Adjusted Prevalence Ratio with 95% CI |
|---|---|---|---|---|
| | No | Yes | | |
| Stunted | 1090 | 1932 | 1.35 (1.21, 1.50) | 1.09 (1.04, 1.13)** |
| **Underweight** | | | | |
| Normal | 2416 | 3283 | 1 | 1 |
| Underweight | 688 | 1408 | 1.51 (1.34, 1.70) | 1.07 (1.03, 1.13)* |
| **Wasting status** | | | | |
| Normal | 2833 | 4059 | 1 | 1 |
| Wasted | 271 | 632 | 1.17 (1.12, 1.23) | 1.03 (0.98, 1.095) |

*p-value<0.05

**p-value<0.01

**Spatial distribution of anemia among under-five children.** The highest prevalence of anemia among children aged 6–59 months was observed in Afar, Somali, Tigray, Dire Dawa, and east Amhara regions (Fig 2). The spatial distribution of anemia among children aged 6–59 months showed significant spatial variation across the country with a global Moran's I value of 0.089 (p-value<0.01) (Fig 3). The statistically significant hotspot areas of anemia were identified in the central, west, and east Afar, Somali, Dire Dawa, Harari, and northwest Gambella regions. While significant cold spot areas were detected in the northwest SNNPRs, Addis Ababa, Benishangul-gumuz, central and southwest Amhara regions (Fig 4).

**The global ordinary least square regression analysis results.** The OLS model was calibrated to diagnose multicollinearity among the independent variables and the mean VIF was less than 10. In the OLS analysis, the model explained about 26% (adjusted $R^2 = 0.22$) of the variation in anemia among children aged 6–59 months with AICc = -169.06. The Joint F-statistics and Wald statistics were significant (p<0.05), which proves that the model was statistically significant. The spatial distribution of residuals was normally distributed as the Jarque-Bera statistics were non-significant (the residuals were normally distributed) (p = 0.18). The Koenker statistics were statistically significant, indicates that the relationship between the independent variables and the dependent variable was non-stationary or heterogeneous across the study areas. This indicates that GWR should be applied (since the Koenker statistics showed the non-stationarity in the relationship) as it assumes the spatial heterogeneity of the relationship between independent and dependent variables across space. The proportion of women aged 15–19 years, the proportion of women who were anemic, the proportion of women who attained formal education, and the proportion of children aged 6–23 months were significantly associated with the percentage of anemia among children aged 6–59 months in the OLS model (Table 3).

**The geographically weighted regression analysis result.** The GWR analysis showed that there was a significant improvement over the global model (OLS). The AICc value decreased from -169.06 (for the OLS model) to -257.41 (for the GWR model). The difference was 88.35 implied that the GWR best explains the spatial heterogeneity of anemia among children aged 6–59 months. Besides, the adjusted $R^2$ was 0.38, the model's ability to explain anemia among children aged 6–59 months has been improved by using GWR since the adjusted $R^2$ was 0.38, indicates that GWR improved the model explaining the power of the OLS model by about 12% (Table 4). In the geographically weighted regression analysis, the proportion of women who had formal education, the proportion of women aged 15–19 years, the proportion of women with poverty, the proportion of being birth order above 6th, proportion of women who had

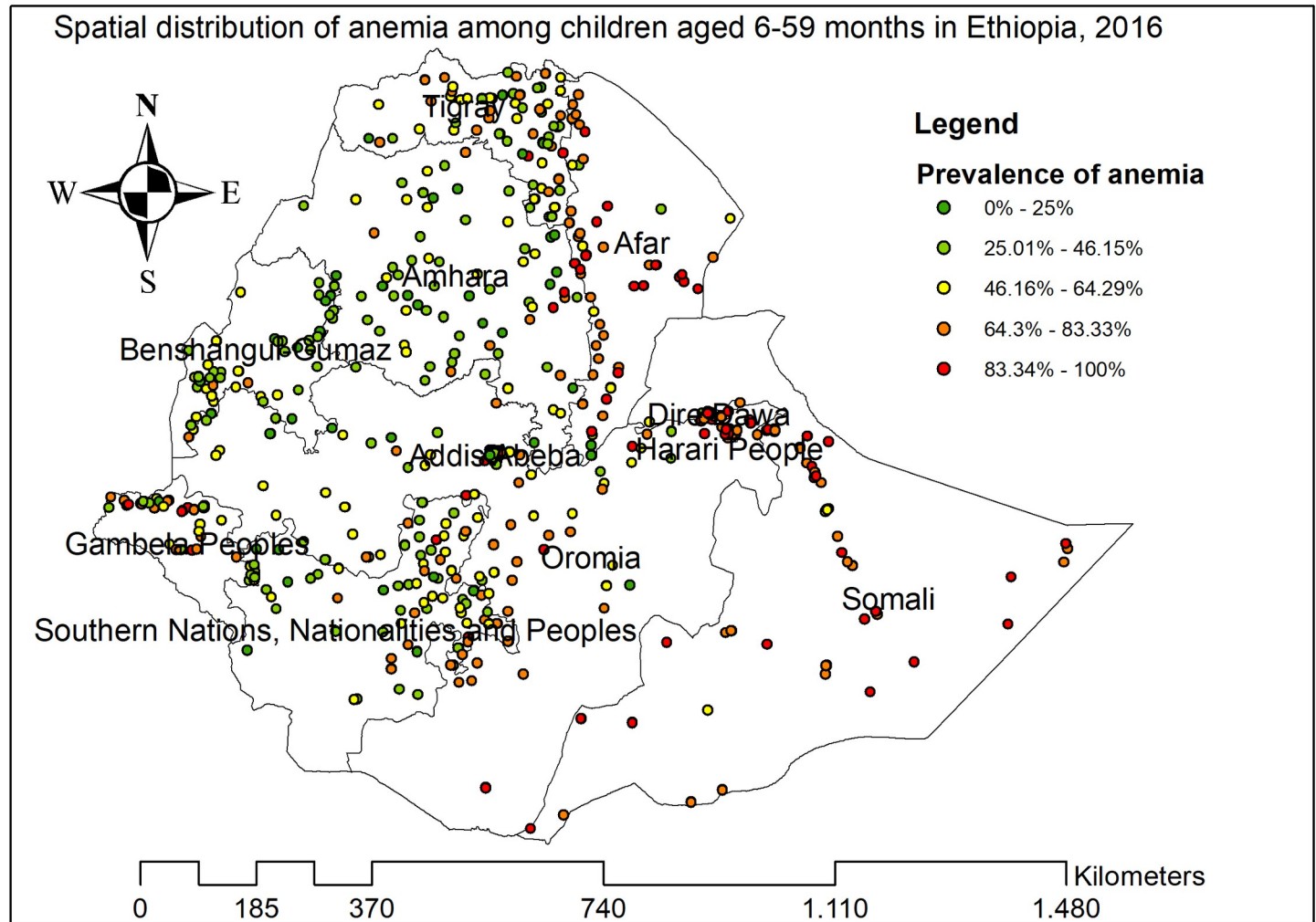

**Fig 2. The spatial distribution of anemia among children aged 6–59 months in Ethiopia, 2016.**

anemia, proportion of children aged 6–23 months, the proportion of stunted children and proportion of wasted children were considered as explanatory variables in the GWR model since it was significant in the multilevel robust Poisson regression analysis as well as have good $R^2$ in the exploratory analysis.

The proportion of women who had anemia had a positive relationship with the proportion of anemia among children aged 6–59 months. As the proportion of women who had anemia increased, the percentage of anemia among under-five children increased in the entire Amhara, Benishangul-gumuz, Gambella, and SNNP regions. The geographic area with red-colored points indicates the highest coefficient of the proportion of maternal anemia (Fig 5). The proportion of mothers aged 15–49 years was significantly associated with the increased risk of anemia among children aged 6–59 months, with the highest effect of mothers age observed in southeast Amhara, east Tire, west Afar, Gambella, and southwest SNNP regions (Fig 6). The proportion of mothers in the poorest household wealth status showed strong and positively associated with increased risk of anemia among under-five children in Somali regions (Fig 7). The proportion of children aged 6–23 months had a significant positive association with anemia among children aged 6–59 months (Fig 8).

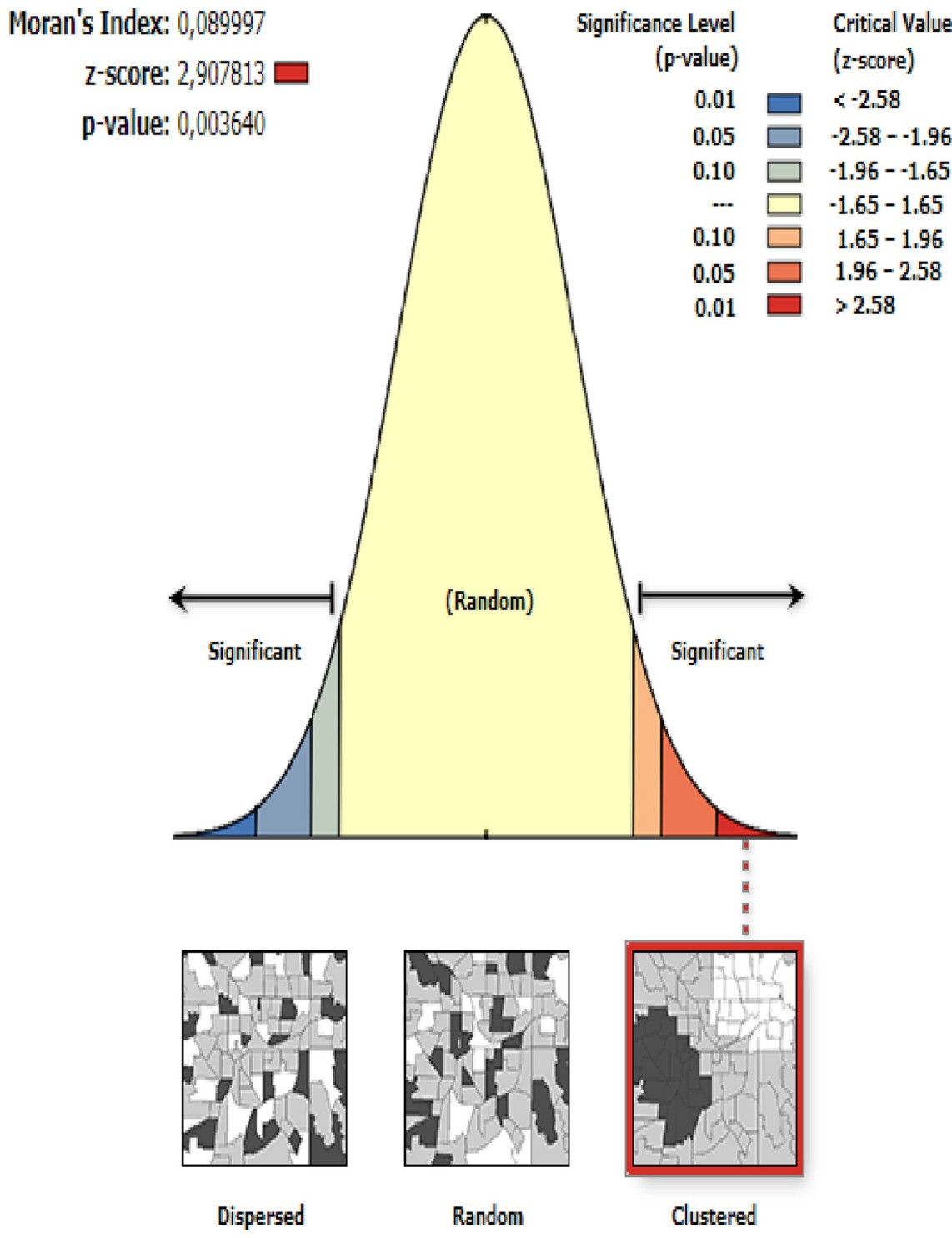

Given the z-score of 2.90781260849, there is a less than 1% likelihood that this clustered pattern could be the result of random chance.

**Fig 3. The global spatial autocorrelation analysis of anemia among children aged 6–59 months in Ethiopia, 2016.**

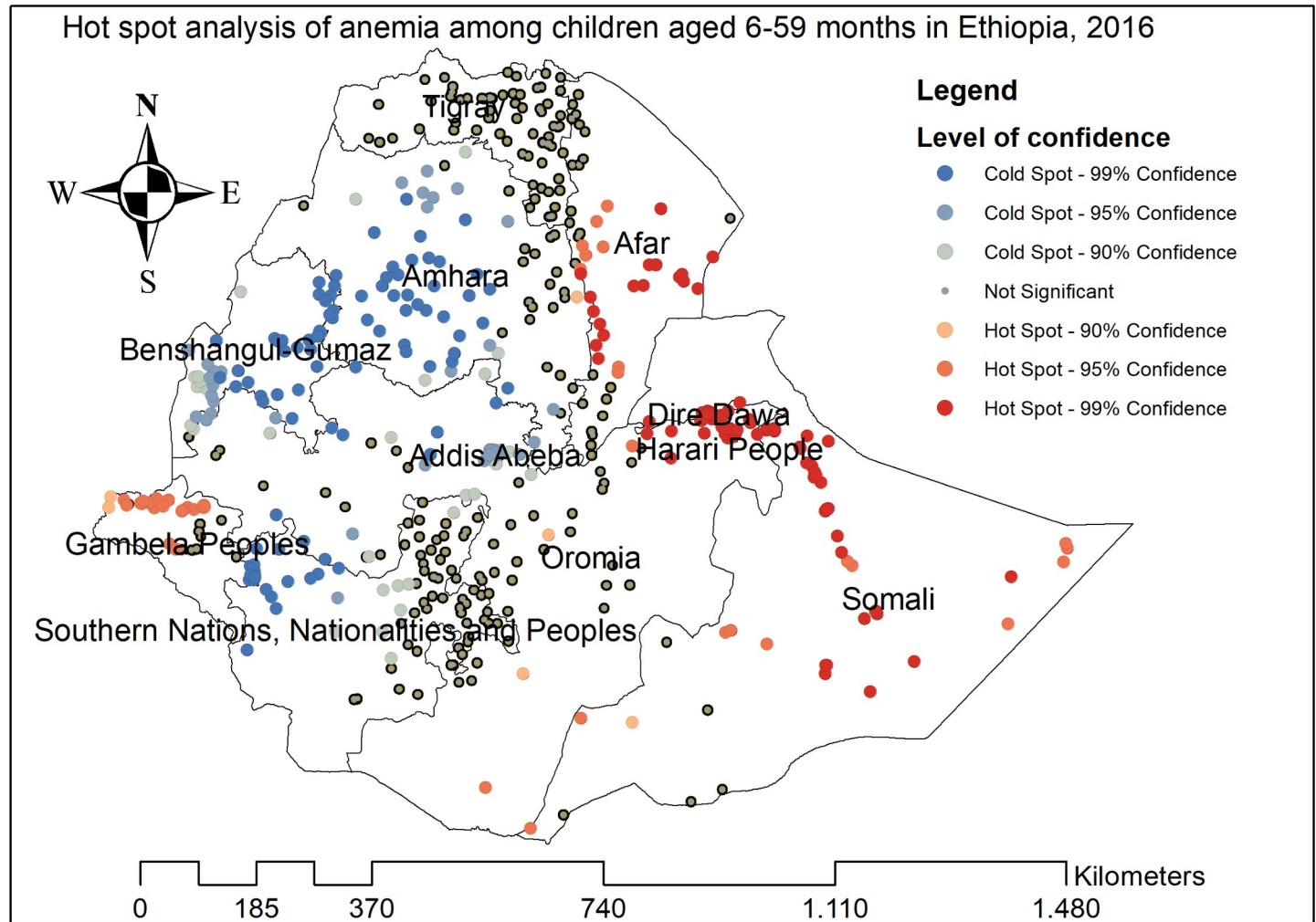

**Fig 4. The Getis Ord Gi statistical analysis of hot spots of anemia among children aged 6–59 months in Ethiopia, 2016.**

## Discussion

Anemia among children aged 6–59 months remains a common public health problem in Ethiopia. In this study, the prevalence of anemia among children aged 6–59 months in Ethiopia was 57.56% (95%CI: 56.50%, 58.61%) ranged from 42.66% in the Amhara region to 83.24% in the Somali region. This was higher than a study reported in Ghana [41] and China [42]. Even though the combined strategies particularly iron supplementation and infectious disease management (such as malaria and helminths infections) are being introduced by the WHO to combat anemia, anemia remains a serious health care problem in Ethiopia. Besides, the spatial distribution of anemia among children aged 6–59 months in Ethiopia was non-random and the hotspot areas of anemia were identified in the central, west, and east Afar, Somali, Dire-Dawa, Harari, and northwest Gambella regions. Though numerous interventions have been implemented to reduce anemia among children such as nutritional interventions like food fortification and supplementation [43–45], iron-folate supplementation during pregnancy [46, 47], appropriate breastfeeding practice [48], and improving personal hygiene [49], there was high spatial heterogeneity of anemia across areas.

**Table 3. The Ordinary Least Square (OLS) regression analysis result.**

| Variable | Coefficient | Robust std-error | Robust t-statistics | Robust probability | VIF |
|---|---|---|---|---|---|
| Intercept | 0.29 | 0.049 | 5.91 | 0.00001* | —— |
| Proportion of mothers aged 15–19 years | 0.27 | 0.12 | 2.21 | 0.027* | 1.09 |
| Proportion of mothers who had formal education | -0.01 | 0.039 | -0.26 | 0.79 | 2.13 |
| Proportion of women with poverty | 0.09 | 0.03 | 2.93 | 0.003* | 1.97 |
| Proportion of birth order greater than 6 | 0.04 | 0.046 | -0.89 | 0.37 | 1.60 |
| Proportion of mothers with anemia | 0.37 | 0.04 | 8.93 | 0.000001* | 1.22 |
| Proportion of children aged 6–23 months | 0.30 | 0.068 | 4.34 | 0.00002* | 1.06 |
| Proportion of stunted children | 0.07 | 0.048 | 1.41 | 0.15 | 1.22 |
| Proportion of wasted children | 0.11 | 0.098 | 1.11 | 0.27 | 1.15 |
| **Ordinary least square regression Diagnostics** | | | | | |
| Number of observations | 615 | Adjusted R-squared | | 0.26 | |
| Joint F-statistics | 27.79 | Prob(>F), (8,600) degree of freedom | | <0.001 | |
| Joint Wald statistics | 221.17 | Prob (> chi-squared), (8) degree of freedom | | < 0.001 | |
| Koenker (BP) statistics | 54.94 | Prob (> chi-squared), (8) degree of freedom | | < 0.001 | |
| Jarque–Bera | 3.38 | Prob (> chi-squared), (2) degree of freedom | | 0.18 | |

VIF: Variance Inflation Factor

The potential reason may be due to the long-standing prevalence of severe malnutrition among under-five children, because of insufficient dietary intake of nutrients in Ethiopia [50, 51]. Besides, Ethiopian children's are highly affected by infectious diseases such as malaria, hookworms, Schistosoma, and visceral leishmaniasis, due to their frequent exposure to poor sanitation and environmental conditions that favor the transmission and spread of parasites [52–54]. In the spatial regression analysis, maternal age, child age, poverty, and maternal anemia was significant predictors of hotspot areas of anemia among under-five children. There is a positive relationship between poverty and hotspots of anemia in Somali regions. Inadequate wealth in the community is associated with an increased risk of intestinal infections such as hookworm, amoebiasis, and ascariasis and these could increase the risk of anemia [55, 56]. Besides, children in the poorest household are prone to undernutrition like lack of folate, iron vitamin B12, and vitamin A [57]. An increased proportion of mothers aged 15–19 years increases the odds of anemia among under-five children in southeast Amhara, east Tigray, west Afar, Gambella, and southwest SNNP regions. The possible explanation might be due to adolescent pregnant mothers are more likely to give low birth weight, preterm or small for gestational age babies [58, 59], in turn, these might contribute to low hemoglobin levels in the blood. The proportion of children aged 6–23 months had a significant positive association with anemia among children aged 6–59 months, this could be due to children aged 6–23 months are at higher risk of exposure to infectious diseases like foreign body aspiration, pneumonia, diarrheal diseases, and intestinal infections because of exposure to complementary feeding, which increases the malabsorption of iron or folate and increases the risk of anemia [60]. In addition, an increased proportion of anemic mothers increases the odds of anemia

**Table 4. Model comparison of OLS and GWR model.**

| Model Comparison parameter | OLS model | GWR Model |
|---|---|---|
| AICc | -169.06 | -257.41 |
| Adjusted R-squared | 0.26 | 0.38 |

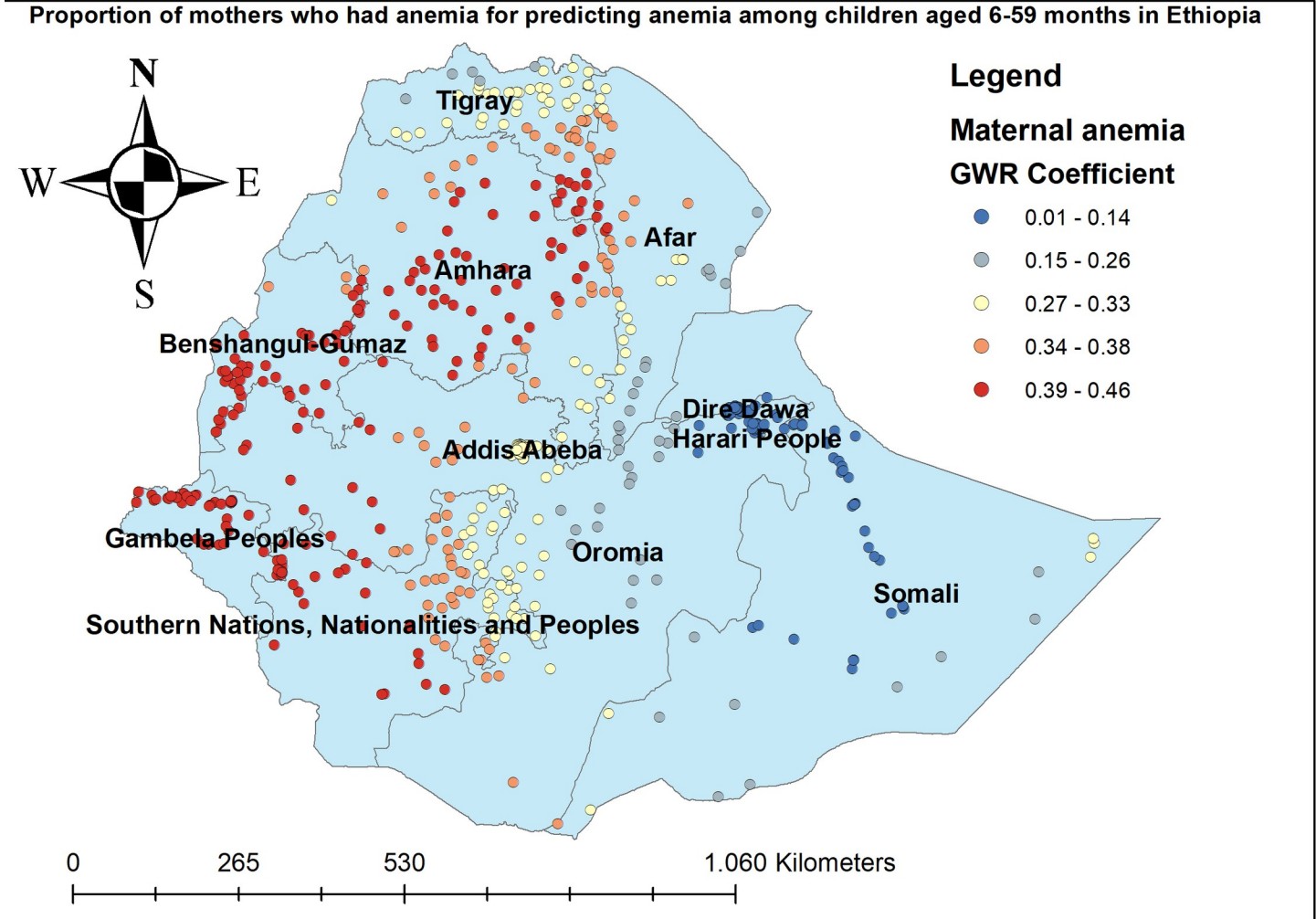

**Fig 5. Mothers with anemia GWR coefficients for predicting anemia among children aged 6–59 months in Ethiopia, 2016.**

among children. This could be due to both mothers and children mostly share a common home environment, which involves mutual exposure to a common set of physical, socioeconomic, and dietary conditions [61]. Also, maternal anemia might be associated with poor birth outcomes such as low birth weight and prematurity of the child, which might lead to limited fetal iron stores and the amount of iron secreted by the breast milk might be insufficient for the daily iron requirement of the child [62].

In the multivariable mixed-effect Poisson regression with a robust variance; maternal age, maternal education, household wealth index, birth order, maternal anemia, age of children, stunting, and underweight were significantly associated with anemia among under-five children.

A child born to a mother aged 30–39 years and 40–49 years had decreased prevalence of anemia compared to a child born to a mother aged less than 20 years. These findings were in line with previous studies reported in Ethiopia [63] and Bangladesh [64]. Anemia is a major public health problem worldwide primarily affects women particularly teenage pregnant girls. Adolescents are particularly susceptible because of their rapid growth and associated high iron requirements. So, pregnancy and lactation increase their nutritional demand, in turn, babies

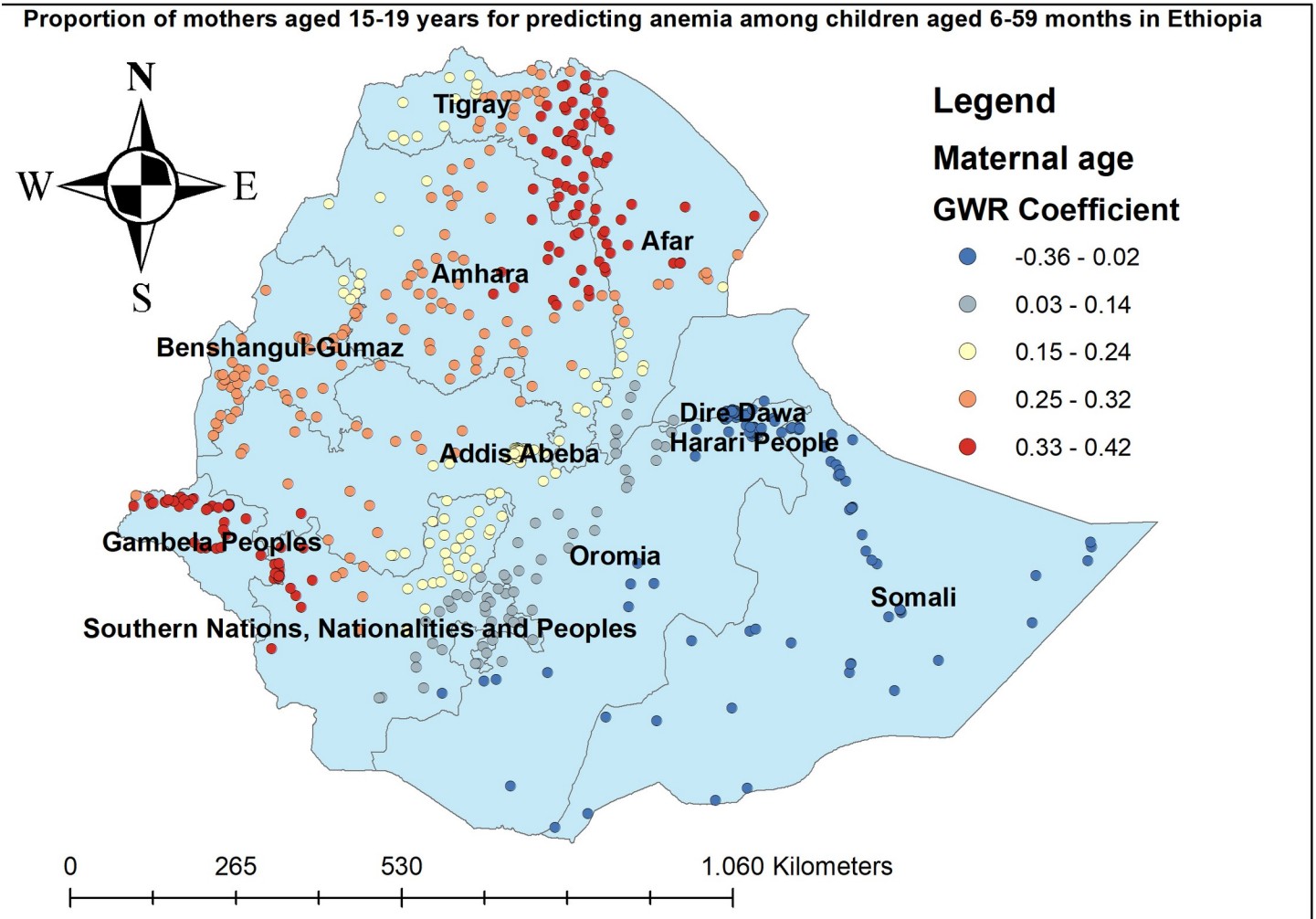

**Fig 6. Mothers aged 15–19 years GWR coefficients for predicting anemia among children aged 6–59 months in Ethiopia, 2016.**

born to teenagers are at higher risk of anemia [65]. Children aged 6–23 months are at higher risk of anemia compared to children aged 24–59 months. This is consistent with study findings reported in Ethiopia [66], and Uganda [67]. It could be due to complementary feeding is initiated after 6 months of birth and during this, a child is exposed to contaminated food and malabsorption syndrome, this could increase nutritional deficiency anemia. In addition, an increased iron requirement due to rapid growth, low availability of foods rich in iron, and lack of diet variety. Iron intake is also likely to improve with age as a result of a more varied diet, including the introduction of meat and other iron-containing foods. The prevalence of anemia was higher among children born to anemic mothers compared to children born to non-anemic mothers. It is in line with study findings in Ghana [68], and Bangladesh [64]. This might be due to children born to anemic mothers are more likely to have nutritional deficiencies like folate, iron, vitamin B12, and vitamin A. Besides, anemic mothers might have underlying diseases such as malaria, HIV/AIDS, or other genetic diseases, and these could transmit to the newborn transplacental, which in turn, could result in anemia. Moreover, mothers and children have common nutritional sources and practices.

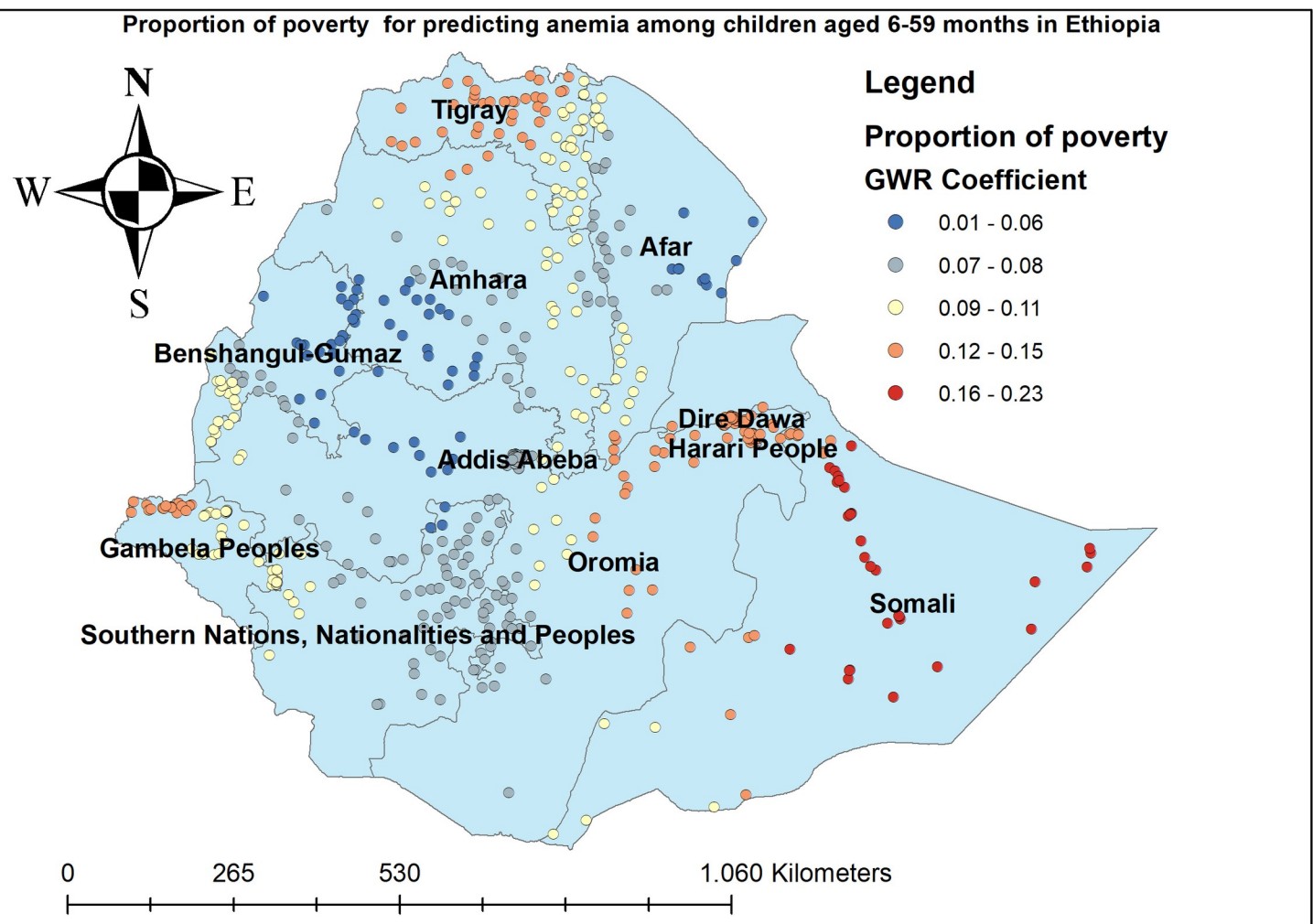

**Fig 7. Child coming from a household with poorer or poorest wealth index GWR coefficients for predicting anemia among children aged 6–59 months in Ethiopia, 2016.**

Another most important significant predictors of anemia among children were maternal education. A child born to a mother who did not have formal education had a higher prevalence of anemia compared to a child born to a mother who attained secondary education or above. It is consistent with study findings reported in Korea [69]. The possible explanation might be due to children born to mothers who did not attain formal education are more likely to consume iron-rich food like meat and poultry compared to children of educated mothers [70]. In addition, educated mothers are more likely to utilize child health services, which can have a positive effect on their children's health outcomes, and improved mothers' level of education results in the corresponding improvement in child feeding practice [71]. Educated mothers exclusively breastfeed their children and initiate appropriate complementary feeding after six months of gestation [72].

High birth order was significantly associated with an increased prevalence of anemia among children aged 6–59 months. This is consistent with studies reported in Sub-Saharan Africa [73] and India [62]. This might be due to as the birth order increases there is maternal

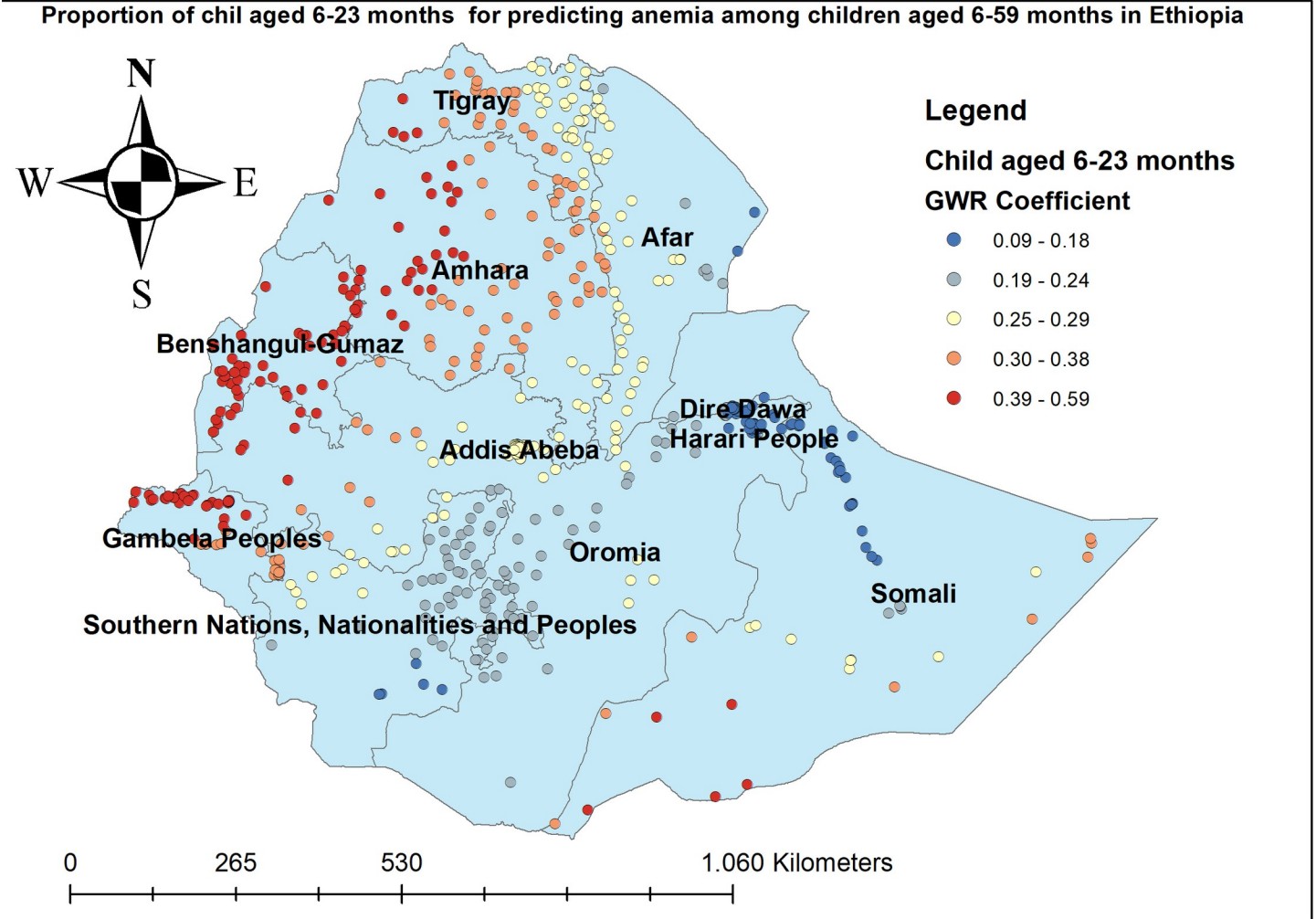

**Fig 8. Children aged 6–23 months GWR coefficients for predicting anemia among children aged 6–59 months in Ethiopia, 2016.**

nutritional depletion and higher-order births are more likely to be low birth weight, this could increase the risk of anemia [74].

In this study, child nutritional status was significantly associated with anemia among children aged 6–59 months. Children who were stunted and/or underweight were more likely to be anemic than their counterparts. This could be due to anemia and malnutrition often share common causes, it is expected that multiple nutrition problems would co-occur in the same individuals. Low intake of iron-rich foods and diminished nutrient absorption caused by changes in the gastrointestinal epithelium in malnourished individuals contribute towards the development of anemia.

## Strength and limitations of the study

This study has several strengths. It is based on the nationally representative DHS data which was weighted and a multilevel model was fitted that enables us to generalize these findings at the national level. Besides, spatial and geographic weighted regression analyses were conducted, the findings can assist the policymakers and program planners to design spatially targeted public health interventions to reduce the incidence of anemia. The findings of this study

should be interpreted considering the following limitations. First, important variables such as underlying medical conditions like malaria, HIV/AIDS, visceral leishmaniasis, and hookworm infections, etc were not considered in the analysis as these variables were not collected in EDHS 2016. Also, we are unable to show the cause-effect relationship between the dependent and independent variables as the DHS data has cross-sectional nature. The geographic locations (GPs) of enumeration areas were displaced up to 2 kilometers in urban and 5 kilometers for most enumeration areas in rural and 10 kilometers for 1% of clusters in rural areas for the sake of privacy, could affect the estimated cluster effects in the spatial regression.

## Conclusion

The prevalence of anemia among children aged 6–59 months in Ethiopia was high, and there was a significant spatial variation of anemia among children aged 6–59 months across regions in Ethiopia. Significant hotspot areas of anemia were identified in the central, west, and east Afar, Somali, Dire Dawa, Harari, and northwest Gambella regions. Being a resident in a geographic area with high community poverty, a high proportion of maternal anemia, a high proportion of children aged 6–23 months, and mothers aged 15–19 years increased the risk of experiencing anemia among children aged 6–59 months. Maternal age, maternal education, birth order, maternal anemia, child age, household wealth index, stunting, and underweight were significant predictors of anemia. Therefore, public health interventions targeting hotspot areas of anemia through empowering women with education help reduce the incidence of anemia among children aged 6–59 months.

## Acknowledgments

We greatly acknowledge MEASURE DHS for granting access to the EDHS data sets.

## Author Contributions

**Conceptualization:** Getayeneh Antehunegn Tesema, Zemenu Tadesse Tessema, Dessie Abebaw Angaw, Koku Sisay Tamirat, Achamyeleh Birhanu Teshale.

**Data curation:** Getayeneh Antehunegn Tesema, Zemenu Tadesse Tessema, Dessie Abebaw Angaw, Koku Sisay Tamirat, Achamyeleh Birhanu Teshale.

**Formal analysis:** Getayeneh Antehunegn Tesema, Zemenu Tadesse Tessema, Dessie Abebaw Angaw, Koku Sisay Tamirat, Achamyeleh Birhanu Teshale.

**Investigation:** Getayeneh Antehunegn Tesema, Zemenu Tadesse Tessema, Dessie Abebaw Angaw, Koku Sisay Tamirat, Achamyeleh Birhanu Teshale.

**Methodology:** Getayeneh Antehunegn Tesema, Zemenu Tadesse Tessema, Dessie Abebaw Angaw, Koku Sisay Tamirat, Achamyeleh Birhanu Teshale.

**Software:** Getayeneh Antehunegn Tesema, Zemenu Tadesse Tessema, Dessie Abebaw Angaw, Koku Sisay Tamirat, Achamyeleh Birhanu Teshale.

**Validation:** Getayeneh Antehunegn Tesema, Zemenu Tadesse Tessema, Dessie Abebaw Angaw, Koku Sisay Tamirat, Achamyeleh Birhanu Teshale.

**Visualization:** Getayeneh Antehunegn Tesema, Zemenu Tadesse Tessema, Dessie Abebaw Angaw, Koku Sisay Tamirat, Achamyeleh Birhanu Teshale.

**Writing – original draft:** Getayeneh Antehunegn Tesema, Zemenu Tadesse Tessema, Dessie Abebaw Angaw, Koku Sisay Tamirat, Achamyeleh Birhanu Teshale.

**Writing – review & editing:** Getayeneh Antehunegn Tesema, Zemenu Tadesse Tessema, Dessie Abebaw Angaw, Koku Sisay Tamirat, Achamyeleh Birhanu Teshale.

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
