## [Decision Letter · Decision Letter 0]

5 Aug 2021

PONE-D-21-12637

Geographic weighted regression analysis of hot spots of anemia and its associated factors among children aged 6-59 months in Ethiopia: A geographic weighted regression analysis and multilevel robust Poisson regression analysis

PLOS ONE

Dear Dr. Tesema,

Thank you for submitting your manuscript to PLOS ONE. After careful consideration, we feel that it has merit but does not fully meet PLOS ONE’s publication criteria as it currently stands. Therefore, we invite you to submit a revised version of the manuscript that addresses all the points raised during the review process.

Particularly, you will see that comments have been made regarding the statistical analysis performed. In addition, it has been suggested that you check the text for the English grammar and syntax.

We look forward to receiving your revised manuscript.

Kind regards,

Francois Blachier, PhD

Academic Editor

PLOS ONE

Journal Requirements:

Reviewers' comments:

Reviewer's Responses to Questions

**Comments to the Author**

1. Is the manuscript technically sound, and do the data support the conclusions?

Reviewer #1: Partly

Reviewer #2: Yes

2. Has the statistical analysis been performed appropriately and rigorously? 

Reviewer #1: No

Reviewer #2: I Don't Know

3. Have the authors made all data underlying the findings in their manuscript fully available?

Reviewer #1: Yes

Reviewer #2: Yes

4. Is the manuscript presented in an intelligible fashion and written in standard English?

Reviewer #1: No

Reviewer #2: Yes

5. Review Comments to the Author

Reviewer #1: The authors present results from a study of anemia among children aged 6-59 months in Ethiopia, its risk factors, and the spatial distribution of cases, based on data from the 2016 Ethiopia Demographic and Health Survey. The manuscript will be strengthened if the authors consider the following points.

1. The authors are encouraged to have the manuscript read by a native English speaker as there are numerous awkwardly phrased sentenced, incomplete sentences, and other grammatical errors. Examples include "varied across regions ranged" (lines 67-68), lines 69-73 (sentence starting with "Half of the" needs to be rephrased), rephrase sentence on lines 81-82, "in every 5 year cycle" should possibly be "every 5 years" (line 92), lines 116-118 (rephrase sentence starting with "Then, they have adjusted"), lines 146-147 ("was done using the Getis-OrdGi* statistics were"), "under-five children non-randomly" (line 156), "Variance Inflation Factor (VIF) values of the VIF" (line 166), "take different value clusters" (line 178), "we employed a mixed-effects Poisson regression with robust variance was fitted" (line 191), and "outcome (anemia) more than 10%" (line 192). These are just some of the phrasing issues, so a careful read-through of the manuscript should be conducted.

2. Authors need to provide more information on the Hgb adjustment as it is not clear when the two parts (separated by "or") of the equation are utilized.

3. The methods of analysis are presented in a different order than the results. For consistency and ease of reading, authors should present the data analysis methods in the same order as the presentation of results, so readers know what to expect.

4. In the spatial regression section of the Methods, authors need to clarify what specifically is being used as the outcome for both the OLS and GWR models. This will help the reader understand the core models as well as the presented results from the models. They also should specify what was used for the bandwidth or neighborhood in the spatial models.

5. Authors go into great detail about the OLS model, including presenting a table with the full results (coefficients) from the model. Yet they state that this model is not appropriate because there is evidence of spatial variability. If that is the case, why present the OLS model?

Minor points:

1. line 61: should "hemoglobulin" be "hemoglobin"?

2. line 185: "STATA" should be "Stata" (https://www.statalist.org/forums/help#spelling)

2. line 132: maybe insert a ":" between "severe anemia" and "for non-pregnant women" and remove "was" from "non-pregnant women was"

3. lines 208-209: authors state that more than 2/3 of the children's mothers were anemic, but Table 1 has that percentage for non-anemic mothers. Authors should correct wherever the error is.

4. Table 1: there is an extra digit in the percentage for no Media exposure.

5. Table 5 is not needed, since the main information is included in the text.

6. line 411: "finings" should be "findings"

7. Figure 3 is not necessary.

8. Title for Figure 8: "chil" should be "children"

Reviewer #2: Background:

Don’t think you need your first sentence in the background. How you define anemia is important for your analysis, but starting with WHO definition isn’t too important.

Most common is quantifiable, what most serious is, is unclear

Again on line 65, does severe mean severity of disease or is it referring to disease frequency

Line 67: are your stats about Ethiopia specifically for children under 5?

Line 69: First sentence doesn’t add much as it’s vague. Are you talking about direct physiologic causes as described in that paragraph or social factors as mentioned in next.

Line 73: Effects of anemia fit better in first paragraph talking about disease burden

Line 81: First sentence of paragraph doesn’t make sense to me.

Are there any studies to cite that do a spatial regression analysis in Ethiopia? Have people does spatial analysis of this issue in other countries (citations?) were the studies useful?

Even if they didn’t use spatial analysis could cite general studies that looked at predictors of anemia in Ethiopia? Specifically call out limitations of these studies to make the need for your study clear

Overall background is clear and makes study purpose clear but could use some edits as mentioned above.

Methods:

Line 99: Were the Kebeles used to create the enumeration areas?

Line 108: What each country’s surveys consist of isn’t too relevant. Instead focus on what Ethiopia’s survey consists of and what you used

Line 114: Important to mention that these cutoffs are based on WHO recommendations and only apply to children under 59 months.

Did you classify anemia by severity? Previous studies have shown differences in mild vs severe

Line 120: Good that you mention values were adjusted for altitude but I don’t think exact formula is too important unless you explain what it means

Line 123: How do you decide which variables to look at? Prior knowledge? If so state that.

Line 152-183: I defer to comments from a biostatistician as this is not my area of expertise

Line 194: What is the rationale for choosing <0.20? What if sometime only wasn’t significant in bivariate analysis due to confounding but you had a strong reason to believe it’s relevant?

Results:

Your factors make sense but I worry that by not sub classifying the severity of anemia in the children you may be missing a certain level of nuance.

Discussion:

You mentioned in your introduction that spatial analysis could be useful for evaluating current interventions and how they are working. It would be helpful if in the discussion you mentioned some of the current interventions and how this relates to your spatial analysis. The way your current discussion section is set up focuses on the predictors of anemia and how they differ regionally but doesn’t really take advantage of the nuance of your analysis.

6. PLOS authors have the option to publish the peer review history of their article (what does this mean?). If published, this will include your full peer review and any attached files.

Reviewer #1: No

Reviewer #2: **Yes: **Luke M Shenton

---

## [Author Response · Author response to Decision Letter 0]

15 Sep 2021

Point by point response for editors/reviewers comments 

Manuscript title: Geographic weighted regression analysis of hot spots of anemia and its associated factors among children aged 6-59 months in Ethiopia: A geographic weighted regression analysis and multilevel robust Poisson regression analysis

Manuscript ID: PONE-D-21-12637

Dear editor/reviewer. 

Dear all,

We would like to thank you for these constructive, building and improvable comments on this manuscript that would improve the substance and content of the manuscript. We considered each comment and clarification questions of editors and reviewers on the manuscript thoroughly. Our point-by-point responses for each comment and questions are described in detail on the following pages. Further, the details of changes were shown by track changes in the supplementary document attached.

 Response to reviewers’ comments 

Reviewer 1

1. The authors present results from a study of anemia among children aged 6-59 months in Ethiopia, its risk factors, and the spatial distribution of cases, based on data from the 2016 Ethiopia Demographic and Health Survey. The manuscript will be strengthened if the authors consider the following points.

Authors’ response: Thank you, reviewer, for your valuable comment. We take all your comments and modified our manuscript extensively. (See the revised manuscript) 

2. The authors are encouraged to have the manuscript read by a native English speaker as there are numerous awkwardly phrased sentenced, incomplete sentences, and other grammatical errors. Examples include "varied across regions ranged" (lines 67-68), lines 69-73 (sentence starting with "Half of the" needs to be rephrased), rephrase sentence on lines 81-82, "in every 5 year cycle" should possibly be "every 5 years" (line 92), lines 116-118 (rephrase sentence starting with "Then, they have adjusted"), lines 146-147 ("was done using the Getis-OrdGi* statistics were"), "under-five children non-randomly" (line 156), "Variance Inflation Factor (VIF) values of the VIF" (line 166), "take different value clusters" (line 178), "we employed a mixed-effects Poisson regression with robust variance was fitted" (line 191), and "outcome (anemia) more than 10%" (line 192). These are just some of the phrasing issues, so a careful read-through of the manuscript should be conducted.

Authors’ response: Thank you so much reviewer. We extensively edited and rewrite the whole manuscript with the support of language experts in the university. We have corrected the grammatical errors, incomplete sentences and typographical errors. (See the revised manuscript)

3. Authors need to provide more information on the Hgb adjustment as it is not clear when the two parts (separated by "or") of the equation are utilized.

Authors’ response: Thank you reviewer. Apologies for the error while we wrote the equation. Currently we modified the equation, and this equation showed how anemia status was assessed among under-five children. As you know altitude was one of the factor that can interfere the risk of anemia and the WHO recommended to correct the cut of points of hemoglobin to define anemia in high altitude populations. Therefore, the EDHS adjusts for altitude to define anemia. (See page 6, line 116, Method section)

4. The methods of analysis are presented in a different order than the results. For consistency and ease of reading, authors should present the data analysis methods in the same order as the presentation of results, so readers know what to expect.

Authors’ response: Thank you for the comments. We have presented the methods of analysis and results in the same order. (See the revised manuscript)

5. In the spatial regression section of the Methods, authors need to clarify what specifically is being used as the outcome for both the OLS and GWR models. This will help the reader understand the core models as well as the presented results from the models. They also should specify what was used for the bandwidth or neighborhood in the spatial models.

Authors’ response: Thank you for the comments. The outcome variable for the spatial regression was the percentage of anemia at the EAs/cluster levels, so, here the study unit is EA/clusters. The explanatory variables were the percentage of the mentioned explanatory variables. Regarding bandwidth, there are three choices of bandwidth methods; AICc, CV and bandwidth parameter. The first two parameters allows to use automatic method for finding the bandwidth which gives the best predictions, whereas bandwidth parameter allows to specify a bandwidth. Considering the merits and demerits of the above alternatives, for this study we have used AICc approach. The AICc method finds a bandwidth which minimizes the AICc values, it is computed from a measure of divergence between the observed and fitted values and a measure of model complexity. (See the revised manuscript, line 166-170, page 8)

6. Authors go into great detail about the OLS model, including presenting a table with the full results (coefficients) from the model. Yet they state that this model is not appropriate because there is evidence of spatial variability. If that is the case, why present the OLS model?

Authors’ response: Thank you for the comments. We presented the OLS results to clearly show for the readers why the GWR model was used and to present the model comparison parameters to compare with GWR. Besides, the results are used for model diagnostics like adjusted R-square, Jarque-Bera, Koenker statistics, and AICc. (See the revised manuscript)

7. Minor points:

1. line 61: should "hemoglobulin" be "hemoglobin"?

Authors’ response: Thank you for the comment. We have addressed it.

2. line 185: "STATA" should be "Stata" (https://www.statalist.org/forums/help#spelling)

Authors’ response: Thank you for the comments. We have addressed it.

3. line 132: maybe insert a ":" between "severe anemia" and "for non-pregnant women" and remove "was" from "non-pregnant women was"

Authors’ response: Thank you for the comments. We have addressed it.

4. lines 208-209: authors state that more than 2/3 of the children's mothers were anemic, but Table 1 has that percentage for non-anemic mothers. Authors should correct wherever the error is.

Authors’ response: Thank you for the comment. We have modified it.

5. Table 1: there is an extra digit in the percentage for no Media exposure.

Authors’ response: Thank you for the comments. We have addressed it.

6. Table 5 is not needed, since the main information is included in the text.

Authors ‘response: Thank you for the comments. We removed it.

7. line 411: "finings" should be "findings"

Authors’ response: Thank you for the comments. We addressed it.

8. Figure 3 is not necessary.

Authors’ response: Thank you for the comments. We preserve this figure because it showed whether the global spatial distribution of anemia is random, dispersed or clustered. Like z-scores, p-value and Moran’s I values. 

9. Title for Figure 8: "chil" should be "children"

Authors’ response: Thank you for the comments. We modified it.

Reviewer#2

1. Don’t think you need your first sentence in the background. How you define anemia is important for your analysis, but starting with WHO definition isn’t too important. Most common is quantifiable, what most serious is, is unclear. Again on line 65, does severe mean severity of disease or is it referring to disease frequency

Authors’ response: Thank you for the comments. We rewrite Background section of the manuscript based on your recommendation. (See the revised manuscript)

2. Line 67: are your stats about Ethiopia specifically for children under 5? Line 69: First sentence doesn’t add much as it’s vague. Are you talking about direct physiologic causes as described in that paragraph or social factors as mentioned in next. Line 73: Effects of anemia fit better in first paragraph talking about disease burden. Line 81: First sentence of paragraph doesn’t make sense to me. Are there any studies to cite that do a spatial regression analysis in Ethiopia? Have people does spatial analysis of this issue in other countries (citations?) were the studies useful? Even if they didn’t use spatial analysis could cite general studies that looked at predictors of anemia in Ethiopia? Specifically call out limitations of these studies to make the need for your study clear. Overall background is clear and makes study purpose clear but could use some edits as mentioned above

Authors’ response: Thank you for the comments. We extensively addressed your comments. The figure we have reported on the prevalence of anemia in Ethiopia is for children under-five. Besides, we wrote about the prevalence of anemia reported in different areas of Ethiopia with appropriate citation and as you can see the prevalence is different. (See the revised manuscript)

3. Methods:

Line 99: Were the Kebeles used to create the enumeration areas?

Line 108: What each country’s surveys consist of isn’t too relevant. Instead focus on what Ethiopia’s survey consists of and what you used

Authors’ response: Thank you for the comments. The EAs were not kebele’s rather EA contains on average 180 households and it is somewhat narrow than kebeles. The EDHS has a number of data sets like men, child, birth, individual, hosehold etc and for this study we have used the kids record file.

4. Line 114: Important to mention that these cutoffs are based on WHO recommendations and only apply to children under 59 months. Did you classify anemia by severity? Previous studies have shown differences in mild vs severe. Line 120: Good that you mention values were adjusted for altitude but I don’t think exact formula is too important unless you explain what it means. Line 123: How do you decide which variables to look at? Prior knowledge? If so state that.

Authors’ response: Thank you for the comments. The cutoff points we used to define anemia was based on WHO which was adjusted for altitude. We have categorized as anemic and non-anemic than severity levels of anemia because we have too few observations in the severe anemic and moderate groups and when we fit a ordinal model the observations are too few and did not fulfilled the assumptions. Besides, we have done the spatial analysis and we aimed to explore anemia either mild, moderate or severe. Variables were selected based on previous literatures and in order to prevent model overfittness we used variables with p<0.2 in the bi-variable analysis and for clinically important variables we included regardless of the p-values.

5. Line 152-183: I defer to comments from a biostatistician as this is not my area of expertise

Line 194: What is the rationale for choosing <0.20? What if sometime only wasn’t significant in bivariate analysis due to confounding but you had a strong reason to believe it’s relevant?

Authors’ response: Thank you for the comments. Here at the beginning we extract variables based on previous literatures and then as we have too many variables, we screened variables for the final model using p-value<0.2 in the bi-variable analysis as a cut of points. Besides, we have seen the LLR values whether adding a variable could improves the model. For clinically important variables we consider in the model regardless of its p-value, and we removed variables which have multicollinearity in the model for keeping model parsimonious and stability. 

6. Results:

Your factors make sense but I worry that by not sub classifying the severity of anemia in the children you may be missing a certain level of nuance.

Authors’ response: Thank you for the comments. We have not categorize the anemia based on the severity of anemia because the prevalence of severe and moderate anemia was too small, majority of the children had mild anemia. Besides, our aim particularly to know the spatial distribution of anemia that is the burden of anemia. That is why we focus on the prevalence of anemia than the severity levels of anemia.

7. Discussion:

You mentioned in your introduction that spatial analysis could be useful for evaluating current interventions and how they are working. It would be helpful if in the discussion you mentioned some of the current interventions and how this relates to your spatial analysis. The way your current discussion section is set up focuses on the predictors of anemia and how they differ regionally but doesn’t really take advantage of the nuance of your analysis.

Authors’ response: Thank for the comments. We incorporated in the discussion sections of the manuscript. (See the revised manuscript)

---

## [Decision Letter · Decision Letter 1]

14 Oct 2021

Geographic weighted regression analysis of hot spots of anemia and its associated factors among children aged 6-59 months in Ethiopia: A geographic weighted regression analysis and multilevel robust Poisson regression analysis

PONE-D-21-12637R1

Dear Dr. Tesema,

We’re pleased to inform you that your manuscript has been judged scientifically suitable for publication and will be formally accepted for publication once it meets all outstanding technical requirements.

Kind regards,

Francois Blachier, PhD

Academic Editor

PLOS ONE

Additional Editor Comments (optional):

Reviewers' comments:

Reviewer's Responses to Questions

**Comments to the Author**

1. If the authors have adequately addressed your comments raised in a previous round of review and you feel that this manuscript is now acceptable for publication, you may indicate that here to bypass the “Comments to the Author” section, enter your conflict of interest statement in the “Confidential to Editor” section, and submit your "Accept" recommendation.

Reviewer #1: All comments have been addressed

2. Is the manuscript technically sound, and do the data support the conclusions?

Reviewer #1: (No Response)

3. Has the statistical analysis been performed appropriately and rigorously? 

Reviewer #1: (No Response)

4. Have the authors made all data underlying the findings in their manuscript fully available?

Reviewer #1: (No Response)

5. Is the manuscript presented in an intelligible fashion and written in standard English?

Reviewer #1: (No Response)

6. Review Comments to the Author

Reviewer #1: (No Response)

7. PLOS authors have the option to publish the peer review history of their article (what does this mean?). If published, this will include your full peer review and any attached files.

Reviewer #1: No

---

## [Editor Report · Acceptance letter]

18 Oct 2021

PONE-D-21-12637R1 

Geographic weighted regression analysis of hot spots of anemia and its associated factors among children aged 6-59 months in Ethiopia: A geographic weighted regression analysis and multilevel robust Poisson regression analysis 

Dear Dr. Tesema:

I'm pleased to inform you that your manuscript has been deemed suitable for publication in PLOS ONE. Congratulations! Your manuscript is now with our production department. 

Kind regards, 

on behalf of

Dr. Francois Blachier 

Academic Editor

PLOS ONE